# The Sweetgum Inscriber, *Acanthotomicus suncei* (Coleoptera: Curculionidae: Scolytinae) Reared on Artificial Diets and American Sweetgum Logs

**DOI:** 10.3390/insects14020186

**Published:** 2023-02-14

**Authors:** Yan Zhang, Xueting Sun, You Li, Lei Gao

**Affiliations:** 1Department of Biology, College of Life and Environmental Sciences, Shanghai Normal University, Shanghai 200234, China; 2Shanghai Academy of Landscape Architecture Science and Planning, Key Laboratory of National Forestry and Grassland Administration on Ecological Landscaping of Challenging Urban Sites, Shanghai 200232, China; 3Vector-Borne Virus Research Center, Fujian Province Key Laboratory of Plant Virology, Fujian Agriculture and Forestry University, Fuzhou 350002, China

**Keywords:** biological characteristics, development, feeding insect, mass rearing, pest

## Abstract

**Simple Summary:**

The sweetgum inscriber *Acanthotomicus suncei* is a newly described pest of American sweetgum in China, which could cause extensive damage to the native flora if this beetle becomes established in North America. Research on the beetle has been hampered by a dwindling access to breeding material. Therefore, the aim of this study was to discover a cost-effective way to rear *A. suncei.* An artificial diet with small amounts of additives supported the shortest development time, whereas rearing the beetles on natural American sweetgum logs led to a longer development time. The lengths and weights of adults reared on the artificial diet were markedly higher than those of adults reared on American sweetgum logs. The egg hatching and eclosion rates of *A. suncei* reared on the artificial diet were significantly higher than those on sweetgum logs. However, the pupation rate of *A. suncei* on the artificial diet was much lower than that on sweetgum logs. In summary, our artificial diet supports a rapid buildup of a healthy *A. suncei* lab population.

**Abstract:**

The sweetgum inscriber, *Acanthotomicus suncei* (Coleoptera: Curculionidae: Scolytinae), is a recently discovered pest of American sweetgum planted in China, with a potential for causing a devastating invasion into North America. Research on the beetle has been hampered by a dwindling access to breeding material. We tested the effect of four artificial diets on *A. suncei’s* developmental time, length and weight of adults, egg hatching rate, pupation rate, and eclosion rate. Additionally, we evaluated the same parameters on *A. suncei* reared on American sweetgum logs. Only one diet supported the full development of *A. suncei* after 30 d. Beetles reared on this diet, which was made of small quantities of agar and additives (i.e., inositol, potassium sorbate, and methylparaben), supported the shortest developmental time of 45.55 ± 1.24 d. Beetles reared on American sweetgum logs exhibited a longer developmental time of 59.52 ± 4.52 d. Beetles reared on the artificial diet were markedly bigger and heavier than those reared on American sweetgum logs (*p* < 0.01). The egg hatching rate (58.90% ± 6.80%) and eclosion rate (86.50% ± 4.69%) of *A. suncei* on the artificial diet were significantly greater than those on sweetgum logs. However, the pupation rate (38.60% ± 8.36%) was much lower on the artificial diet than on the sweetgum logs. Here, we reported the best artificial diet for *A. suncei* and discuss the advantages and disadvantages over rearing the beetle on American sweetgum logs.

## 1. Introduction

Bark beetle (Curculionidae: Scolytinae) outbreaks are devastating to managed forests and planted trees throughout the world. Some of the outbreaks are caused by climate-related tree stress and the homogenization of stands, such as the outbreaks of conifer bark beetles that affected millions of hectares of trees in North America and Eurasia [1,2]. In other cases, invasive bark beetle species are introduced by humans through trade into regions with new host trees that lack resistance and cause damage to these native, naïve trees [3,4,5]. To prevent, monitor, and manage future bark beetle invasions, it is imperative that species with invasion potential are studied in the country of their origin before potential invasion.

The sweetgum inscriber, *Acanthotomicus suncei* (Curculionidae: Scolytinae: Ipini) was proved to be a deadly pest to American sweetgum, *Liquidambar styraciflua,* and a secondary colonizer of the Chinese sweetgum, *L. formosana,* in China, causing a slight mortality rate of Chinese sweetgum [6,7]. In eastern China where American sweetgum was planted extensively, the beetle caused outbreaks that effectively eliminated this imported tree from the landscape and nursery market [6,8]. Given the potential losses to nursery stock and urban areas, it is imperative that researchers find ways to detect and control this beetle to prevent its potential establishment and massive damage in North America, where the American sweetgum is a common tree [8,9].

Breeding the sweetgum inscriber in the laboratory is crucial for research and the development of pest detection and control methods. Laboratory bioassays are necessary for testing pesticides or pheromones and the development of new treatments requires the cost-effective mass production of hundreds or thousands of individuals at each stage of the pest’s development [10]. Much of a bark beetle’s life is spent in the larval stage concealed beneath tree bark, making access and observation difficult. This can be alleviated by using an artificial diet [11,12]. Scolytine bark beetles have been reared in the laboratory from the egg to the adult stage on natural diets [13,14,15,16,17] and artificial diets [18,19,20,21,22,23,24,25].

The effect of an artificial diet on insect development needs to be validated and compared to its fitness on a natural log. A high-quality artificial diet greatly facilitates research. To advance the laboratory research on *A. suncei*, we developed and tested an artificial diet for rearing larvae. In this study, we present the technique and compare the biological parameters of populations reared on American sweetgum logs and the artificial diet.

## 2. Materials and Methods

### 2.1. Insect Rearing

The experiments were conducted at the Laboratory of Plant Protection, Shanghai Academy of Landscape Architecture Science and Planning (SALASP), Shanghai, China. Boles of American sweetgum infested with *A. suncei* were collected from Lingang New Town, Pudong District, Shanghai, China (121.92° E, 30.93° N). The adult beetles of *A. suncei* used in the experiments were obtained from these infested tree boles and stored in plastic containers [26]. Healthy boles of American sweetgums (8~10 cm DBH), which were collected from the Geqichun nursery, Pukou District, Nanjing, Jiangsu province (118.55° E, 32.09° N), were cut into 20 cm segments, sent to SALASP, and sealed with wax and paraffin to prevent water evaporation.

Fresh logs were mixed with infested logs, and when new adults started emerging, newly colonized segments (with signs of boring dust and entrance holes) were removed and placed vertically in plastic containers (21 × 12 × 11 cm) with 200 mesh gauze and kept in incubators under controlled temperature and humidity conditions, and the date was recorded. Two circular holes with a diameter of 7 cm were made on the opposite sides of each container. Twelve replicate containers were kept in incubators (MIR 350H, Sanyo Electric Co., Ltd., Osaka, Japan) with 25 ± 1 °C, 70% relative humidity (RH), and complete darkness.

Four formulas of artificial diets were tested for rearing larvae of *A. suncei* (Table 1). The phloem powder was obtained from healthy American sweetgum which was ground and sieved through 0.9 mm mesh and then dried in an oven at 72 °C for 48 h. The compositions were formulated based on previous research on other bark beetle species [19,27].

The yeast powder and sucrose were mixed in 25 mL of distilled water in a beaker while heated in a hot water bath at 52 °C and stirred until dissolved. Then the agar, inositol, Wechsler salt, multivitamin, cholesterol, ascorbic acid, potassium sorbate, and methylparaben were dissolved, adding 15 mL of distilled water in the beaker under the same condition. The wheat germ powder, microcrystalline cellulose, bark and phloem powder, and 10 mL of distilled water were then added to the solution and stirred. The mixture was fluffy in the beaker. The final mixture in the beaker was wrapped, sealed, and stored at 4 °C in the refrigerator for the beetle rearing test. The artificial diet was not solid in the fridge.

Using a chisel, eggs were collected from excavated galleries in American sweetgum logs. Eggs were transferred gently to a sheet of black filter paper in glass Petri dishes (100 × 20 mm) using a moist sable brush (size 000). In total, 394 eggs were collected for the artificial diet test. In an attempt to minimize contamination from the original galleries, all eggs were soaked in 75% alcohol for 10 s in glass Petri dishes (100 × 20 mm) using a sable brush (size 000), and rinsed with sterile water in glass Petri dishes for 10 s. All eggs were sterilized one by one. They were individually placed at the bottom of 1.5 mL Eppendorf tubes which were then filled with 1 mL of the artificial diet for the growth of *A. suncei* larvae. When the artificial diet was heated to 25 °C, the artificial diet could be used for rearing. We used a spoon to fill Eppendorf tubes with artificial diet and squeezed the artificial diet with a glass stirring rod. Once the tubes were filled, they were all kept under the same conditions and left until the adults emerged.

### 2.2. Growth and Development of A. suncei

In the natural host plant treatment, the number of adults in each container was counted and left undisturbed until adult emergence. The time between the boring of the parental adults and the emergence of their offspring was recorded and used to calculate the average generation development duration (GDD). The calculation was performed as follows:$$\bar{x}=\left(x_{1}f_{1}+x_{2}f_{2}+x_{3}f_{3}+\cdots +x_{k}f_{k}\right)/\overset{k}{\underset{1}{\sum }}\;f_{i}\bar{x}=\left(x_{1}f_{1}+x_{2}f_{2}+x_{3}f_{3}+\cdots +x_{k}f_{k}\right)/\overset{k}{\underset{1}{\sum }}\;f_{k}$$

In the formula, *x*_1,2,3,…,*k*_ represent the duration of the emergence of offspring adults from when the parents initially bored on the 1st, 2nd, 3rd...*k* day; *f*_1,2,3,…,*k*_ represent the number of offspring adults collected on the 1st, 2nd, 3rd...*k* day.

Upon the emergence of all offspring adults, the infested bark was removed with chisels and tweezers, and the numbers of egg traces, sub tunnels, and pupa cells in each gallery were counted and recorded [28]. The rates of hatching, pupation, and eclosion were calculated for each gallery as follows:$$Hatching\;rate=\left(\frac{number\;of\;sub\;tunnels}{number\;of\;egg\;traces}\right)\times 100\%$$
$$Pupation\;rate=\left(\;\frac{number\;of\;pupa\;cells}{number\;of\;sub\;tunnels}\right)\times 100\%$$
$$Eclosion\;rate=\left(\frac{number\;of\;adults}{number\;of\;pupa\;cells}\right)\times 100\%$$

In the artificial diet treatment, the generation development duration was determined in the same manner as the natural host plant treatment. The head widths of larvae were measured on the 5th, 10th, 15th, and 20th days after egg hatching and the color morphological characteristics were observed. The egg hatching, pupation, and eclosion rates of *A. suncei* were calculated using the following formulae:$$Hatching\;rate=\left(\frac{number\;of\;larvae}{number\;of\;eggs}\right)\times 100\%$$
$$Pupation\;rate=\left(\frac{number\;of\;pupae}{number\;of\;larvae}\right)\times 100\%$$
$$Eclosion\;rate=\left(\frac{number\;of\;adults}{number\;of\;pupae}\right)\times 100\%$$

The offspring adults were separated by sex based on morphological characteristics under stereomicroscope [6], and the parameters of their body weights (accurate to 0.1 mg) and body lengths (accurate to 0.1 mm) were measured.

### 2.3. Statistical Analysis

The data was analyzed using GraphPad Prism 8 software. The homogeneity of variance was tested for all data, and the body weights of larvae raised on artificial diets were analyzed using ordinary one-way ANOVA. The body weights and body lengths of male and female adults reared on artificial diets and American sweetgum logs were compared and analyzed using independent sample *t*-tests.

## 3. Results

### 3.1. Optimized Artificial Diet

The results showed that diet B was the optimal formulation for rearing *A. suncei* from larvae to adult (Table 2). The biological parameters of *A. suncei* reared on artificial diets were then analyzed based on diet B.

### 3.2. Growth Characteristics

The body color of the younger larvae is light brown. The color of mature larvae changes from light brown to milky white and develops bristles distributed all over its body (Figure 1). When *A. suncei* entered the pupation stage, they either formed a chamber inside the diet or burrowed out onto the diet’s surface to pupate. During the eclosion, the color of their eyes changed from white to black and their black eyes gradually developed first, followed by the development of their mouth, elytra, and pronotum (Figure 2).

The lengths and weights of male and female adults reared on the artificial diet were significantly higher compared to the American sweetgum logs treatment (*p* < 0.01), as shown in Figure 3. The differences in egg hatching rate, pupation rate, and eclosion rate of *A. suncei* reared in American sweetgum logs and the artificial diet were significant (*p* < 0.01), as shown in Table 3. The egg hatching rate (58.09% ± 6.80%) and eclosion rate (86.50% ± 4.69%) of *A. suncei* on the artificial diet were significantly higher than those on sweetgum logs with a hatching rate of 33.87% ± 7.32% and an eclosion rate of 58.05% ± 4.84%. The pupation rate of *A. suncei* larvae in the American sweetgum logs (69.40% ± 8.63%) was higher than larvae reared on the artificial diet (38.60% ± 8.36%) (Table 3).

### 3.3. Developmental Time

Under the artificial diet treatment, there were three larval instars of *A. suncei*, and head widths were measured every 5 days. The head widths of the larvae were measured on the 5th, 10th, 15th, and 20th day and were 0.36 ± 0.02 mm, 0.47 ± 0.01 mm, 0.50 ± 0.01 mm, and 0.50 ± 0.01 mm, respectively (Figure 4). The head widths increased rapidly between the 5th to 15th day (*p* < 0.01), but there was no significant difference after the 15th day (*p* > 0.05). The average generation developmental time was 45.44 ± 1.24 days on the artificial diet, which is significantly shorter than 59.52 ± 4.52 days on American sweetgum logs (Figure 5).

## 4. Discussion

In the study, diet B was found to be the best formulation for rearing *A. suncei* from larva to adult. The use of agar in the diet helped maintain its hardness and water content, which were important for the feeding of insects [29]. However, a high proportion of agar in diet C led to 100% mortality after 20 days. The high mortality in diet A may be attributed to the production of toxins by the preservatives [30]. Adding inositol to diet B may have contributed to its success, as previous studies [31] showed that inositol is needed for the development of larvae into adults in other bark beetle species.

Our study demonstrates that *A. suncei* can successfully undergo its larval and pupal stages when reared on the newly designed artificial diet B. Furthermore, our results indicate that *A. suncei* can develop from the egg stage to adulthood without the need for changing the diet, a deviation from the rearing methods used for other scolytine beetles, such as *Dendroctonus armandi* [32] and *Xyleborus pfeili* [33]. Our newly designed artificial diet for *A. suncei* remained stable from the egg stage to adulthood.

The use of Eppendorf tubes as containers for beetle rearing is cost-efficient as it requires minimal amounts of the artificial diet. Additionally, this method enables the observation of specific behaviors, such as gallery construction, within the artificial diet in live insects [34,35,36]. Our results showed that the developmental duration of *A. suncei* was significantly influenced by different food sources. Development was faster and adult weight and length were greater when reared on the artificial diet compared to American sweetgum logs. This difference may be attributed to the more readily accessible nutrients in the artificial diet. Among the artificial diets evaluated, artificial diet B was the best as it had the highest survival rate and shortest developmental time, likely due to its nutrition content [37]. The addition of cacao bark extract to the cellulose-based artificial diet for the ambrosia beetle *Xyleborus ferrugineus* (Fabricius) was found to accelerate gallery construction initiation compared to other diets [38].

The egg hatching and eclosion rates of *A. suncei* reared on the artificial diet were significantly higher compared to those reared on American sweetgum logs. However, the pupation rate of *A. suncei* fed on American sweetgum logs was significantly higher than those fed on the artificial diet. This disparity may be attributed to factors such as media humidity or chamber texture, or to the presence of fungi and parasites on the natural logs. Eggs on the artificial diet were sterilized to reduce contamination in the study, whereas eggs on the natural logs may have been affected by the poor nutritional quality and allelochemicals present in the sweetgum logs [39,40,41]. Bark beetles require sterols for eclosion [42], and the addition of sterols in future artificial diet formulations may further improve the development of *A. suncei*.

The use of artificial diets has significant impacts on bark beetle and ambrosia beetle research. Artificial diet systems are useful in evaluating the potential bio-insecticide effects of fungal lectins and their potential use in pest control. The recent invasive cryptic ambrosia beetle, *Euwallacea fornicatus* Eichhoff, has been reared on artificial diets to study its biology and sex ratios [43]. Adult males and females of *Ips typographus* were fed an artificial diet treated with various concentrations of compounds, which resulted in reduced feeding in proportion to the concentration. Males were found to be more responsive to antifeedants at higher concentrations compared to females [44].

In conclusion, our study shows that the artificial diet is a suitable substitute for American sweetgum logs for the development of *A. suncei*. The use of the diet in laboratory bioassays can aid in research on control options for the beetle, such as pesticides, pheromones, and antibiotics. However, further improvement is needed in the rearing process and apparatus for large-scale, multi-generation breeding in laboratory conditions.

## Figures and Tables

**Figure 1 insects-14-00186-f001:**
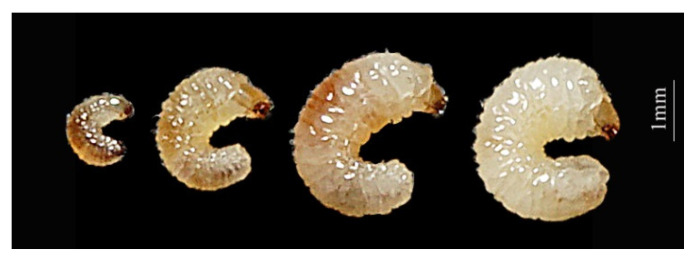
Morphology of *Acanthotomicus suncei* larvae from 5th to 20th day (left to right: 5 d, 10 d, 15 d, 20 d).

**Figure 2 insects-14-00186-f002:**
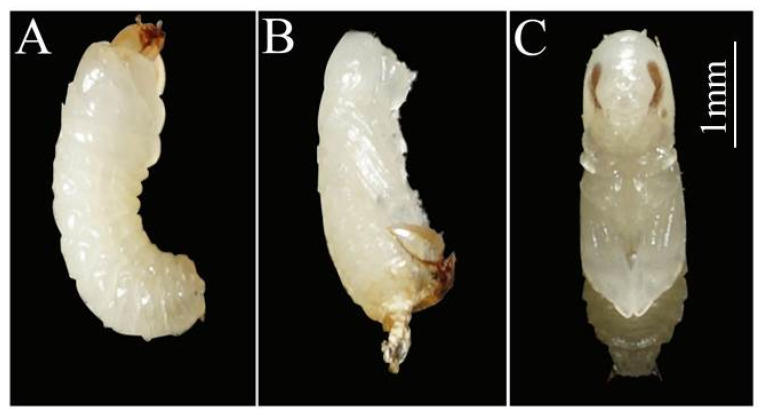
Morphological changes of *Acanthotomicus suncei* from larva to pupa; (**A**) mature larva; (**B**) callow pupa (the shell shed to the bottom); (**C**) mature pupa with the black eyes gradually developed.

**Figure 3 insects-14-00186-f003:**
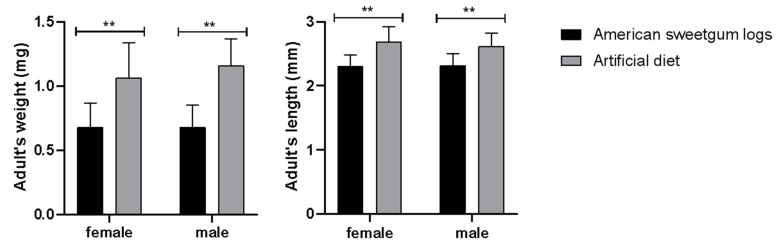
Differences in weight and length between males and females of *Acanthotomicus suncei* reared on the artificial diet and American sweetgum logs. Mean ± SD, ** *p* < 0.01.

**Figure 4 insects-14-00186-f004:**
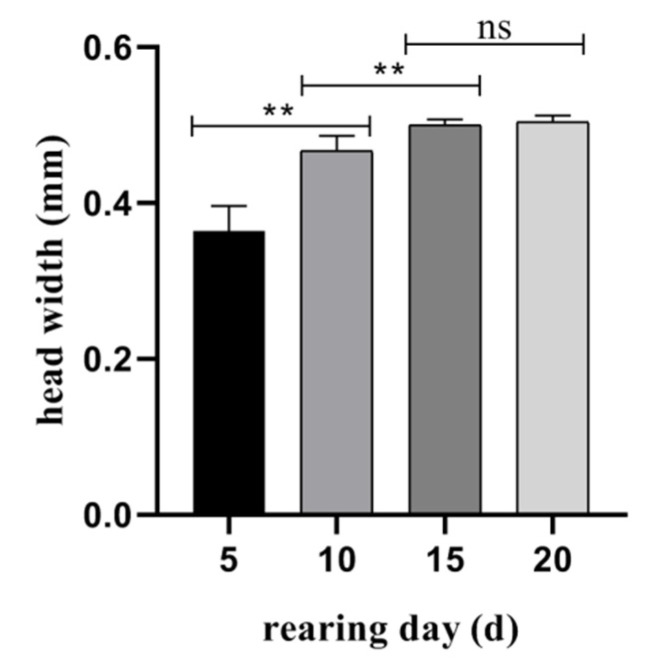
Differences in larval head width of *Acanthotomicus suncei* reared on the artificial diet. Mean ± SD, ** *p* < 0.01, ns = not significant.

**Figure 5 insects-14-00186-f005:**
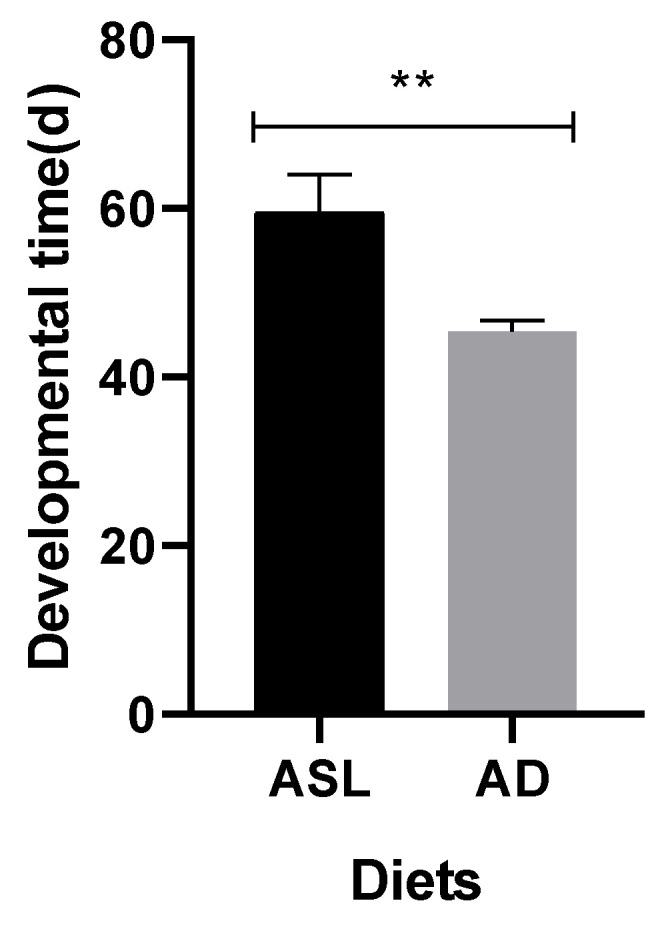
Developmental time (d) of *Acanthotomicus suncei* reared on American sweetgum logs and the artificial diet. ASL: American sweetgum logs; AD: artificial diet. Mean ± SD, ** *p* < 0.01.

**Table 1 insects-14-00186-t001:** Formulas of artificial diets for rearing the larvae of *Acanthotomicus suncei*.

Composition	A	B	C	D
sweetgum tree phloem powder/g	17.50	17.50	17.50	17.50
wheat germ powder/g	3.75	3.75	3.75	3.75
agar/g	3.00	3.00	6.00	3.00
microcrystalline cellulose/g	11.50	11.50	11.50	11.50
yeast powder/g	5.00	5.00	5.00	5.00
sucrose/g	2.50	2.50	2.50	2.50
inositol/g	0.10	0.10	0.10	-
Wechsler salt/g	0.10	0.10	0.10	0.10
multivitamin/g	0.25	0.25	0.25	0.25
cholesterol/g	0.50	0.50	0.50	0.50
ascorbic acid/g	0.10	0.10	0.10	0.10
potassium sorbate/g	0.10	0.10	0.10	0.10
methylparaben/g	0.30	0.10	0.10	0.10
distilled water/mL	50.00	50.00	50.00	50.00

**Table 2 insects-14-00186-t002:** Survival rate of *Acanthotomicus suncei* fed with artificial diets recorded every five days.

Artificial Diets	Survival Rate						
	1 d	5 d	10 d	15 d	20 d	25 d	30 d
A	100%	60%	22%	12%	0%	0%	0%
B	100%	66%	52%	48%	38	34%	22%
C	100%	62%	30%	20%	14%	0%	0%
D	100%	60%	34%	24%	18%	2%	0%

**Table 3 insects-14-00186-t003:** Comparison of egg hatching rate, pupation rate, and eclosion rate of *Acanthotomicus suncei* reared on American sweetgum logs and the artificial diet.

Treatment	Egg Hatching Rate	Pupation Rate	Eclosion Rate
artificial diet	58.90% ± 6.80% ^a^	38.60% ± 8.36% ^b^	86.50% ± 4.69% ^a^
sweetgum logs	33.87% ± 7.32% ^b^	69.40% ± 8.63% ^a^	58.05% ± 4.84% ^b^

Mean ± SD, within columns, means followed by different letters differ significantly at *p* < 0.05.

## Data Availability

Data will be shared upon request to the corresponding author.

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
