# Peer review of "The Sweetgum Inscriber, Acanthotomicus suncei (Coleoptera: Curculionidae: Scolytinae) Reared on Artificial Diets and American Sweetgum Logs"

_insects, 2023, doi:10.3390/insects14020186_

Round 1

Reviewer 1 Report

The authors demonstrated that the new artificial diet which they have developed are suitable for rearing of a serious pest and potential invader, sweetgum bark beetle, Acanthotomicus suncei. The design of the experiment and the analysis of the data are correct. The results of the present study is certainly important for any further laboratory work on this insect and therefore the manuscript can be published, although it needs a number of minor corrections and improvements (see below).

Lines 18 delete “and” before “egg hatching”.

Line 12: replace “dramatically higher” with “markedly higher”.

Line 31: delete “and” before “egg hatching”.

Lines 33-34: delete “despite the limited pupation rate” (data on pupation rate are given below, lines 38-39)

Line 35: replace “dramatically higher” with “markedly higher”.

Line 51: replace [93-6] with [3-6].

Lines 82-83: “A. suncei” should be in Italics.

Line 116: replace “in the same circumstance” with “under the same conditions”.

Line 148: replace “stereoscope” with “stereomicroscope”.

Lines 173, 174, etc. Please, indicate what descriptive statistics is used in your paper: mean and SEM or mean and SD, or something else.

Figure 1: Please, explain the meaning of the two asterisks in the legend (I guess, p<0.01).

Please, indicate what is shown: mean and SEM or mean and SD, or something else.

Figure 3: Please, explain the meaning of the two asterisks in the legend.

Please, indicate what is shown: mean and SEM or mean and SD, or something else.

Line 202: “Acanthotomicus suncei” should be in Italics.

Line 204: replace “within each data set” with “within each column”

Line 207: “A. suncei” should be in Italics.

Line 209: replace “of development” with “days of development”.

Figure 4: Please, explain the meaning of the two asterisks and “ns” in the legend.

Please, indicate what is shown: mean and SEM or mean and SD, or something else.

Line 234: replace “dramatically higher” with “markedly higher”.

Finally, please, carefully check the format of the references: year should be in Bold font (line 281), journal name should start with capital letters (line 284), etc. 

Author Response

Dear reviewer

I would like to take this great opportunity to thank you for valuable comments. We have addressed all the comments carefully and the revised portions are highlighted in red in the manuscript. I believe that the addressing of all these comments has greatly improved the quality of this manuscript.

Point 1: Lines 18 delete “and” before “egg hatching”.

Response 1: We are really sorry for our careless mistakes. Thank you for your reminder. We have deleted “and” before “egg hatching”. Please see the revised portions are highlighted in red in the manuscript.

Point 2: Line 22: replace “dramatically higher” with “markedly higher”.

Response 2: We have replaced “dramatically higher” with “markedly higher”. Thank you for your suggestions. Please see the revised portions are highlighted in red in the manuscript.

Point 3: Line 31: delete “and” before “egg hatching”.

Response 3: We have deleted “and” before “egg hatching”. Please see the revised portions are highlighted in red in the manuscript.

Point 4: Lines 33-34: delete “despite the limited pupation rate” (data on pupation rate are given below, lines 38-39)

Response 4: We have deleted “despite the limited pupation rate” in Lines 25

Point 5: Line 35: replace “dramatically higher” with “markedly higher”.

Response 5: We have replaced “dramatically higher” with “markedly higher”. Please see the revised portions are highlighted in red in the manuscript.

Point 6: Line 51: replace [93-6] with [3-6].

Response 6: Sorry for our careless mistakes. We have replaced [93-6] with [3-6]. (line 52)

Point 7: Lines 82-83: “A. suncei” should be in Italics.

Response 7: Thanks for your careful checks. We have corrected “A. suncei” to “A. suncei”.

Point 8: Line 116: replace “in the same circumstance” with “under the same conditions”.

Response 8: Thank you for your suggestions. We have replaced “in the same circumstance” with “under the same conditions”. (line 139)

Point 9: Line 148: replace “stereoscope” with “stereomicroscope”. 

Response 9: We have replaced “stereoscope” with “stereomicroscope”. (line 159)

Point 10: Lines 173, 174, etc. Please, indicate what descriptive statistics is used in your paper: mean and SEM or mean and SD, or something else.

Response 10: We have added mean and SD in our paper. Thank you so much for your careful checks. Please see the revised portions are highlighted in red in the manuscript.

Point 11: Figure 1: Please, explain the meaning of the two asterisks in the legend (I guess, p<0.01). Please, indicate what is shown: mean and SEM or mean and SD, or something else.

Response 11: We have explained the meaning of the two asterisks in the legend. We also have added mean and SD below the Figure 1. Please see the revised portions are highlighted in red in the manuscript.

Point 12: Figure 3: Please, explain the meaning of the two asterisks in the legend. Please, indicate what is shown: mean and SEM or mean and SD, or something else.

Response 12: We have explained the meaning of the two asterisks in the legend. We also have added mean and SD below the Figure 3. Please see the revised portions are highlighted in red in the manuscript.

Point 13: Line 202: “Acanthotomicus suncei” should be in Italics.

Response 13: Thanks for your careful checks. We have corrected “Acanthotomicus suncei” to “Acanthotomicus suncei”. Please see the revised portions are highlighted in red in the manuscript.

Point 14: Line 204: replace “within each data set” with “within each column”.

Response 14: We have replaced “within each data set” with “within each column”. Please see the revised portions are highlighted in red in the manuscript.

Point 15: Line 207: “A. suncei” should be in Italics.

Response 15: Thanks for your careful checks. We have corrected “A. suncei” to “A. suncei”. Please see the revised portions are highlighted in red in the manuscript.

Point 16: Line 209: replace “of development” with “days of development”.

Response 16: We have replaced “of development” with “days of development”.

Point 17: Figure 4: Please, explain the meaning of the two asterisks and “ns” in the legend. Please, indicate what is shown: mean and SEM or mean and SD, or something else.

Response 17: We have explained the meaning of the two asterisks in the legend. We also have added mean and SD below the Figure 4. Please see the revised portions are highlighted in red in the manuscript.

Point 18: Line 234: replace “dramatically higher” with “markedly higher”.

Response 18: We have replaced “dramatically higher” with “markedly higher”. Please see the revised portions are highlighted in red in the manuscript.

Point 19: Finally, please, carefully check the format of the references: year should be in Bold font (line 281), journal name should start with capital letters (line 284), etc. 

Response 19: We have carefully checked the format of the references according to your suggestions. Please see the revised portions are highlighted in red in the manuscript.

We appreciate for reviewer’s warm work earnestly, and hope the correction will meet with approval. Once again, thank you very much for your comments and suggestions.

Reviewer 2 Report

Interesting manuscript about the sweetgum inscriber beetle Acanthotomicus suncei. However, the manuscript still needs an extensive revision. Several suggestions are provided below.

In general:

Simple summary and abstract: Both (simple summary and abstract) are almost identical. What is the point of a simple summary if the authors just repeat the abstract? Please see more details below.

Introduction: Easy to ready and follow. It states the (potential) problem and provides enough details of why the study was needed. Some minor comments were provided.

Materials and methods: Several comments/suggestions are given. Please remember to provide enough details so your experiment(s) can be replicated by others.

Results: This section needs to be partially re-organized following the suggestions provided. Please avoid jumping back and forth among subjects. Please consider the following order: (1) describing the best artificial diet, (2) describe the color and morphology of larvae and pupae, (3) developmental time of A. suncei, which include head capsules, length, and weight of individuals.

Discussion: This section is the one that needs more work. As it is, it’s just repetition of the results, some interesting ideas are provided but lack support. It does not flow well. More details are provided below.

Specific comments:

L2-4:      is this manuscript mainly focused on the biology of the beetle or the use of artificial diet? I believe the main point is the artificial diet. Please consider the following title: The sweetgum inscriber beetle, (Acanthotomicus suncei Cognato, Coleoptera: Curculionidae: Scolytinae) reared in artificial diet and American sweetgum logs.

L14-26: The idea of a simple summary is to convey the most important message (i.e., the take-home message) to a non-scientific community. It should be written in a simple way, without technical jargon, without numbers, percentages, and p-values. For the simple summary considering just using just the common name. The way it is currently written is just a copy/past of the abstract. Please, see other simple summaries as examples.

L27-40: Please consider the following abstract: “The sweetgum inscriber Acanthotomicus suncei (Coleoptera: Curculionidae: Scolytinae) is a recently discovered pest of American sweetgum planted in China, with a potential for devastating invasion into North America. Research of the beetle has been hampered by decreasing access to breeding material. We tested the effect of four artificial diets on A. suncei developmental time, length and weigh of adults, egg hatching, pupation, and eclosion rate. Additionally, we evaluated the same parameters on A. suncei reared on American sweetgum logs. Only one diet supported the full development of A. suncei after 30 d. Beetles reared in this diet, which was comprised of small quantities of agar, and additives (i.e., inositol, potassium sorbate, and methylparaben) supported the shortest developmental time, 45.55 ± 1.24 d. Beetles reared on American sweetgum logs exhibited a longer (59.52 ± 4.52 d) developmental time. Beetles reared on artificial diet were bigger and heavier than those reared on American sweetgum logs (P<0.01). The egg hatching rate (58.09 ± 6.80%) and eclosion rate (86.50 ± 4.69%) of A. suncei on artificial diet were significantly greater than those on sweetgum logs. However, the pupation rate (38.60 ± 8.36%) was much less on artificial diet than on sweetgum logs. Here, we report the best artificial diet for A. suncei and discuss the advantages and disadvantages over rearing the beetle on American sweetgum logs”.

L41:        Keywords should be in alphabetical order; delete “biological characteristics”. Please use words that can increase the visibility of your manuscript.

L51:        [93-6], there is no reference #93. Please correct.

L61:        Please consider: “its potential establishment”

L64-65:  Please consider: “Laboratory bioassays are required for tests of pesticides, pheromone research, or the effect of antibiotics”.

                Additionally, what are the authors implying by “the effect of antibiotics”? Has this beetle been associated with a pathogenic bacterium? Would it be possible the authors mean the effect of antibiotics on the artificial diet? I don’t see how “the effect of antibiotics” is relevant here. Please either elaborate or delete.

L81:        Replace “Acanthotomicus suncei” with “A. suncei

L82-83:  A. suncei should be italicized

L84:        Delete “the”; 8~10 cm? DBH

L88-89:  “…containing hundreds of families with larvae or pupae” These are a bunch of words that do not add anything to the paragraph. Please avoid the use of unnecessary words. Please consider the following: “Freshly cut healthy logs of sweetgum were mixed with A. suncei infested logs”

L89-91: “When new adults started emerging and boring into the new logs, each newly colonized segment (with signs of boring dust and entrance holes) was removed, and the date was recorded”

                Can the authors provide the number of beetles per colonized segment? Did they just assume it was one beetle per segment?

L92:        Please include space between symbols. For example: 21 × 12 × 11 cm

L94:        You should never start a sentence with “actual” numbers. Please, replace “12” for “Twelve”

L97-99:  “The compositions were selected from various formulations previously used for bark beetles [20,28,29]”

                The above sentence is misleading. Reference #20 used rearing media for bark beetles (Ips). However, why do the authors cite references #28 (grain moth larvae) and 29 (predators and parasitoids) as bark beetles? Do these references cite bark beetles?

L105:      Interventionary. What is the purpose of this word? Is it a typo? a subheading? Why is it there? is this word really necessary for the document? Please avoid the use of unnecessary words.

L106-107: “The yeast powder and sucrose were mixed in 25 mL distilled water while heated at 106 °C in a water bath and stirred until dissolved”.

                Did the authors used a Beaker or an Erlenmeyer for this? Once dissolved, were the rest of ingredients added to the yeast powder and sucrose mixed? What happened with the other 25 mL of distilled water? Is it possible that the rest of the ingredients homogenized separately (in the other 25 mL) and then mixed with the yeast and sucrose? Please remember to provide enough details for the experiment to be replicated by others.

L108-109: “preservative”, are the authors talking about the potassium sorbate and methylparaben? It would be better to use the actual names instead of the word “preservative”.

L110-111: “The final solution was wrapped, sealed and stored at 4°C in the refrigerator for the beetle rearing test”

                HOW exactly was the “solution” stored? Since the authors used agar, and the agar solidifies after cooling, then it is not technically a solution. Did the authors pour the solution and let it solidify in a different container? Perhaps the diet never solidified? According to L219 the artificial media got solidified.

L112:      “In total, 394 eggs were collected for the artificial diet test”

                HOW were eggs collected? Did the author excavate the galleries from infested logs? Where did they (authors) get the eggs from? Did you establish a colony of the beetle in the artificial diet first, and later harvested those eggs?

L113:      “all eggs were soaked in 75% alcohol for 10s, and rinsed with sterile water for 10s”

                Again, HOW? Bark and ambrosia beetles are a couple of mm long, therefore, eggs are minute!!! Did the authors just got a “bunch of eggs” and submerged them in EtOH? I want to know how did you “sterilized” these eggs. Again, please provide enough details so others can replicate your experiment.

L114:      Please replace “separately” with “individually”; replace “in” for “at”

L115:      “transparent centrifuge tube”. Why use three words when two can do? I am assuming the authors used Eppendorf tubes, am I correct?

L115-116: “and then filled with the above artificial diet for the growth of the larvae of A. suncei

                Here are my questions to the authors, did you wait until the eggs hatched and then poured the artificial diet into the tube? Or, assuming the diet got solidified, did you cut a “piece” and introduced it into the tube? This is critical for someone that would like to replicate your experiment.

L117:      “…and no more diet was added until the offspring adults emerged”

                I am confused with the above sentence. Does it mean that, once the beetles emerged, the authors added more diet? Why!?

L118:      Please consider the following subheading title: “Growth and development of A. suncei

L121:     “the number of days from the parental adult’s boring to the emergence of offspring adults”

                Just to confirm here, the authors meant the last emerging offspring adult, right?

L125-127: This entire paragraph is just a simple average. Providing that formula just make it confusing. Why the authors need “the number of adults collected”? aren’t they interested only in developmental time? Would the results change drastically if they remove the number of adults collected? I’m guessing no.

                Average = ∑ x / n, where “x” is the value of each observation (developmental time), and n is the total number of observations (12 replicates).

                For example: Sum of all observations (a1 + a2 + a3 + …… a12) divided by the total number of observations (12 replicates used).

Would the authors please simplify this paragraph, be consider with the reader.

L144:      Please delete the word “infant” from the equation. What does infant imply? It does not add anything new. Please avoid unnecessary words.

L158-159: “It was found that only diet B could feed A. suncei from egg to adult by using four artificial diets (Table 2).”

                The sentence above is poorly written. Technically the eggs do not feed. “by using four artificial diets” this does not make sense. Please rephase.

L158-161: This whole paragraph is poorly written. Does not flow and it is hard to read. Please rephrase.

L170:      Please replace the word “formulas” with “diets”

L171:      “The figures recorded in the table”. Could the authors please use a better word? Thanks.

                Line 171 could be easily included in the table title, please merge them.

L172:      Developmental time

Figure 1: The authors need to add the X-axis label. You just don’t present a figure (graph) without label. Remove the legend and write the treatments in the x-axis.

                Y-axis label should read developmental time; please delete the space in ( d)

L178:      Please consider the following title: “Developmental time (d) of Acanthotomicus suncei reared on American sweetgum logs and artificial diet”

L180:      Please consider a new title for this section, “growth characteristics” does not describe the paragraph. Additionally, if the authors plan to describe part of the life cycle (larvae and pupae) of A. suncei, it would make more sense to do it after 3.1 section.

                The authors can easily merge lines 211-213 with lines 181-184 and have one nice paragraph that describes the immature life stages of A. suncei.

L186:      Please replace “morphology” with “morphological”

L192-195: Please consider the following changes: “The egg hatching rate (58.09 ± 6.80%) and eclosion rate (86.50 ± 4.69%) of A. suncei on artificial diet were significantly higher than the egg hatching rate (33.87 ± 7.32%) and eclosion rate (58.05 ± 4.84%) on sweetgum logs.

L195:      Delete “but”

L198:      WHY does this figure do not have a caption? Please in the Y-axis, include a space after adult’s length.

Figure 3. It seems the figure caption here applies to the previous figure. If that is the case, please group both graphs in one big figure. It is clear the authors have 2 separate figures, but the way it is presented gives the impression that one figure does not have a figure caption.

L202:      The scientific name should be italicized

Table 3. The egg hatching rate on artificial diet is different from the one given in the abstract and result sections. Please fix.

L204:      Please replace “within each data set” with “within each column”.

L207:      A. suncei should be italicized

L209:      During the 5th to 15th day of development

L211-213: Please consider merging these lines with lines 181-184, it would make a better paragraph. L211-213 cut the flow of the paragraph, makes it difficult to read.

L212:      What is “flesh-colored”? Human skin color? What color!? This term is vague, subjective, and politically incorrect. Please address accordingly.

L218:      The discussion section can be greatly improved. Currently, it lacks flow, some sentences are poorly written. The authors present some interesting ideas, but these were not supported with additional information. Some sentences are just repetitions of the results section. For instance, the authors do not talk about the importance of artificial diets in bark and ambrosia beetles. How artificial diets have facilitated the study of biology and fungal associates in bark and ambrosia beetles. A lot of articles have been published recently in this regard. Why using diets can be advantageous over the use of logs.

L225-226: “Artificial diet studies of Scolytus multistratus by Baragao [33] suggested that larvae needed inositol to develop into adults”

                Interesting statement. But HOW does inositol help larvae developing into adults? The authors just throw the idea without really expanding it.

L228-230: “The artificial diets of other scolytine beetles, such as Dendroctonus armandi [34] and Xyleborus pfeili [35] needed to be changed during the rearing process, while our diet of A. suncei survived from egg to adult on the same batch”

                I don’t really see how this is relevant. Many bark and ambrosia beetles can complete their development in a single artificial batch, please see the coffee berry borer, X. crassiusculus, Euwallacea species, and several Xyleborus species.

L231:      “This research obtained larvae with weights on various days (Figure 3)”

                On various days? This is confusing and poorly written. What are the authors trying to say? This needs more explanation for the reader to understand.

L231-232: “The weight gain of mature larvae was more obvious than young larvae on the artificial diet”

                Well, this is sort of a given. Mature individuals (3rd instar larvae) would be heavier than young (1st instar larvae) individuals.

L232-234: “It’s consistent with silkworm [36] which has reported 5th instar larvae grow rapidly and the body mass was dramatically higher than that of young larvae on an artificial diet”

                I understand the authors are trying to compare their results. However, this is like comparing apples and oranges. Yes, both organisms are insects, but they are not even in the same order. Why would you make that type of comparison?

L235-326: “We used a transparent centrifuge tube [37-39] filled with the artificial diet so that the physical changes of A. suncei from egg to adult could be easily observed”

                The sentence above is totally irrelevant. Why are the authors trying to explain the use Eppendorf tubes? I don’t see how this sentence is useful.

L240-243: “Those phenomena may be attributed to more accessible nutrient sources in the artificial diet which were consistent with the results of…..”

                Excellent idea, however, it would be even more interesting if the authors develop the idea. Why are the nutrients more or less accessible to the beetles? Please explain. The whole purpose of a discussion is to provide the reader with potential explanations of your results. Comparing the results with somebody else’s without providing explanations is not scientific. Finally, why again the authors compare their results to lepidopterans such as Spodoptera frugiperda.

L252:      “in the natural logs thank in the diet” Please correct the typo.

L263-266: According to the section “author contributions” only two (YZ and LG) provided most of the work in this manuscript. Please state what other contributions were provided by XS and YL. They way it is right now it seems they were just included in the ms out of courtesy.

L276-368: Please double-check every single reference. Most of them do not match the reference guidelines for the journal.

All scientific names should be italicized; scientific names start with capitalized letters; orders and family names must start with a capitalized letter; the name of all journals must be italicized; authors should use the abbreviation for all journals (please see web of science journal title abbreviations). Please refer to the authors guidelines, section 8.11, ACS reference style.

Author Response

Dear reviewer

Round 2

Reviewer 2 Report

Thanks to the authors for addressing my original comments. The results and discussion sections read better now.

Please find several additional minor comments that will improve the quality of this manuscript.

L40: Please replace “developed” with “reported”

L42: The authors provided a new set of keywords; however, this change is not reflected in the revised manuscript. Please address accordingly.

L63: Please insert space between “tree[8-9]”

L72: Replace “if” with “of”

L72-77: Please use simple spacing

L90: Please move “(with signs of boring dust and entrance holes)” before “were”. It will read much better.

L93-94: Please consider the word “opposite” instead of “symmetrical”. I think it reads better.

L89-96: Please use simple spacing

L99: Please add the word “and” between “mesh then”

L109-118: In the current version, please be advised the line numbers are superposed on the table, which makes it hard to read. If the authors can fix it, please do so.

L125: Thank you for clarifying. The word mixture is way more appropriate. Good job!

L128-131: Please consider the following changes to the text “Using a chisel, eggs were collected from excavated galleries in American sweetgum logs. Eggs were transferred gently to a sheet of black filter paper in glass Petri dishes (100 × 20 mm) using a moist sable brush (size 000).”

L133-134: Since the authors already provided the size of glass Petri dishes and sable brush in the previous lines, they are not necessary in these lines. Please delete the parenthesis and its content.

L135-138: Please reorganized these sentences. The way is currently written suggest that the egg was placed at the bottom and then the artificial diet was pushed in. This would smash the egg. Based on the picture the authors provided, the artificial diet was pushed in first, and then on the surface, they placed the egg. Please address accordingly.

L136: Please consider “for the growth of A. suncei larvae”

L139: Please consider “and left undisturbed until adult emergence”

L150: “was” instead of “were”

L215: Please use “within columns” instead “within data”

L237-238: “Diet A may be attributed to the production of toxins by preservatives which increased mortality”

This does not read well. Please consider the following “High mortality in Diet A may be attributed to the production of toxins by the preservatives”

L305: Xyleborus glabratus should be italicized. Please use capital letters for: eichhoff, insecta, coleoptera, curculionidae, scolytinae

L367: “Xyleborus” instead of “xyleborus”. Please use capital letters for the journal abbreviation.
